# Resveratrol’s Pro-Apoptotic Effects in Cancer Are Mediated Through the Interaction and Oligomerization of the Mitochondrial VDAC1

**DOI:** 10.3390/ijms26093963

**Published:** 2025-04-22

**Authors:** Tal Raviv, Anna Shteinfer-Kuzmine, Meital M. Moyal, Varda Shoshan-Barmatz

**Affiliations:** 1Department of Life Sciences, Ben-Gurion University of the Negev, Beer-Sheva 84105, Israel; talush37@gmail.com; 2National Institute for Biotechnology in the Negev, Ben-Gurion University of the Negev, Beer-Sheva 84105, Israel; shteinfe@post.bgu.ac.il (A.S.-K.); petrescu@post.bgu.ac.il (M.M.M.)

**Keywords:** apoptosis, hexokinase, mitochondria, resveratrol, VDAC1

## Abstract

Resveratrol is a naturally occurring phenolic compound found in various foods such as red wine, chocolate, peanuts, and blueberries. Both in-vitro and in-vivo studies have shown that it has a broad spectrum of pharmacological effects such as providing cellular protection and promoting longevity. These effects include antioxidant, anti-inflammatory, neuroprotective, and anti-viral properties, as well as improvements in cardio-metabolic health and anti-aging benefits. Additionally, resveratrol has demonstrated the ability to induce cell death and inhibit tumor growth across different types and stages of cancer. However, the dual effects of resveratrol—acting to support cell survival in some contexts, while inducing cell death in others—is still not fully understood. In this study, we identify a novel target for resveratrol: the voltage-dependent anion channel 1 (VDAC1), a multi-functional outer mitochondrial membrane protein that plays a key role in regulating both cell survival and death. Our findings show that resveratrol increased VDAC1 expression levels and promoted its oligomerization, leading to apoptotic cell death. Additionally, resveratrol elevated intracellular Ca^2+^ levels and enhanced the production of reactive oxygen species (ROS). Resveratrol also induced the detachment of hexokinase I from VDAC1, a key enzyme in metabolism, and regulating apoptosis. When VDAC1 expression was silenced using specific siRNA, resveratrol-induced cell death was significantly reduced, indicating that VDAC1 is essential for its pro-apoptotic effects. Additionally, both resveratrol and its analog, trans-2,3,5,4′-tetrahydroxystilbene-2-O-glucoside (TSG), directly interacted with purified VDAC1, as revealed by microscale thermophoresis, with similar binding affinities. However, unlike resveratrol, TSG did not induce VDAC1 overexpression or apoptosis. These results demonstrate that resveratrol-induced apoptosis is linked to increased VDAC1 expression and its oligomerization. This positions resveratrol not only as a protective agent, but also as a pro-apoptotic compound. Consequently, resveratrol offers a promising therapeutic approach for cancer, with potentially fewer side effects compared to conventional treatments, due to its natural origins in plants and food products.

## 1. Introduction

Resveratrol (3,4′,5-trihydroxy-trans-stilbene) is a natural compound produced by plants as a phytoalexin and is found in various food sources, including red wine, chocolate, peanuts, and blueberries [1,2,3]. In nature, the compound exists in two geometric isomers: trans and cis, with the trans form being more stable than the cis form [4]. Both isomers are present in wine; however, comprehensive studies on isomerization have shown that trans-resveratrol is primarily responsible for the health benefits associated with the compound [5]. As a stilbene, a subclass of phenolic compounds, resveratrol has potential effects on cell defense mechanisms, making it a popular health supplement [2,3,6,7]. Resveratrol has been demonstrated to offer mitochondrial protection both directly and indirectly, contributing to extended cell life; although, it can also induce mitochondrial dysfunction and trigger apoptosis [8]. Furthermore, resveratrol has been recognized for its pro-apoptotic properties, promoting cell death and acting as an anti-cancer agent [9,10,11,12].

Several naturally occurring and synthetic hydroxylated analogs of resveratrol have also been identified [4]. These common analogs include oxyresveratrol, pterostilbene, trans-2,3,5,4′-tetrahydroxystilbene-2-O-glucopyranoside (TSG), and acetyl-trans-resveratrol [13].

Resveratrol is known for its wide range of beneficial biological activities, including antioxidant, anti-inflammatory, anti-cancer, neuroprotective, and anti-viral properties. It also has been found to lower blood pressure and improve cardio-metabolic markers, along with anti-aging effects [14,15,16,17,18,19]. Due to its ability to protect cells and organisms from various forms of damage, resveratrol is considered a health-promoting compound and is commonly used as a dietary supplement because it offers numerous health benefits [20].

However, resveratrol also exhibits pro-apoptotic activity by downregulating anti-apoptotic proteins that promote cell survival such as Bcl-2, Bcl-xL, survivin, and XIAP, while upregulating pro-apoptotic proteins like Bax, Bak, PUMA, Noxa, P21, Bim, various caspases, and other pro-apoptotic factors [21,22,23]. This results in the induction of cell death and the suppression of cancer growth at various stages, making it a potential alternative treatment for cancer or a complementary therapy alongside chemotherapy [9,10,11,12,24,25,26]. The resveratrol pro- and anti-survival effects on cancer are thought to be mediated via different pathways and targets, but sometimes the same protein or gene undergoes downregulation or upregulation in the presence of resveratrol, depending on its activity in cell protection or cell death [15].

The pro-survival and pro-apoptotic effects of resveratrol on cancer cells are thought to be mediated through distinct pathways and targets, with the same proteins or genes potentially being either downregulated or upregulated, depending on whether the compound is promoting cell protection or cell death [15].

Resveratrol’s proposed mode of action involves a variety of mechanisms and pathways such as activating signal transduction pathways and alerting gene expression [26]. Its key targets include NF-κB, COX-2, and superoxide dismutase 1 (SOD1), which contribute to its anti-inflammatory and antioxidant effects. As a polyphenol, resveratrol can transfer electrons to reactive species, disrupting oxidative cascades and effectively scavenging reactive oxygen species (ROS) [27]. Additionally, it regulates the expression of pro-oxidative and anti-oxidative enzymes, increasing the levels of SOD1, which catalyzes the conversion of superoxide radicals (O_2_^−^) into oxygen (O_2_) and hydrogen peroxide (H_2_O_2_), while also upregulating glutathione peroxidase 1 (GPx1) and downregulating the NADPH oxidase subunit Nox4 [28].

In the context of cancer, resveratrol targets various signaling pathways involved in cancer development and progression, making it a promising candidate for cancer treatment [26]. It activates the phosphatase and PTEN/PKB signaling pathway [29] as well as PTEN, a tumor suppressor gene that is often downregulated in many cancers. Additionally, resveratrol activates other tumor suppressor genes, such as AMP-activated protein kinase (AMPK) and p53, which promote apoptosis and help prevent tumorigenesis. Resveratrol also inhibits the rat sarcoma virus (RAS) and disrupts the associated proliferation pathway, including the Raf/MEK/ERK signaling cascade, contributing to its anti-cancer effects [23,30,31].

As an anti-cancer agent, resveratrol also inhibits the tumor necrosis factor (TNF), a cytokine that lays a role in immune cell activation, differentiation, cell migration, and angiogenesis [32]. Furthermore, resveratrol inhibits inducible nitric oxide synthase (iNOS), which generates nitric oxide (NO), a signaling molecule involved in angiogenesis and considered an inflammatory-related protein [33]. It also plays a role in immune-cell activation, differentiation, cell migration, and angiogenesis [34].

This study presents another target of resveratrol, the mitochondrial protein VDAC1, located in the outer mitochondrial membrane (OMM). VDAC1 acts as a mitochondrial gatekeeper that mediates the transport of ions and metabolites between the mitochondrion and the cytosol [35,36,37]. A key protein in mitochondria-mediated apoptosis [35,36,37,38,39,40], VDAC1 plays a role in releasing mitochondrial pro-apoptotic proteins from the intermembrane space (IMS) into the cytosol and interacts with apoptosis-regulating proteins such as hexokinase (HK), Bcl-2, and Bcl-xL [36,39,40,41,42,43].

Recently, we demonstrated that apoptosis inducers such as cisplatin, selenite, H_2_O_2_, UV light, and various stress conditions lead to VDAC1 overexpression, shifting the protein’s equilibrium towards oligomerization. These oligomeric forms of VDAC1 create large channels that facilitate the release of pro-apoptotic proteins, ultimately triggering cell death [35,36,38,39,41,42,44].

Here, we demonstrate that resveratrol directly interacted with purified VDAC1, enhancing its expression, promoting its oligomerization, and inducing apoptotic cell death. Additionally, resveratrol increased intracellular [Ca^2+^] and ROS levels and induced the detachment of HK from VDAC1.

These findings suggest that resveratrol-mediated apoptosis is linked to increased VDAC1 expression and oligomerization, positioning resveratrol not only as a protective compound, but also as a pro-apoptotic agent that offers a potential therapeutic strategy for cancer.

## 2. Results

### 2.1. Resveratrol Induces VDAC1 Overexpression, Oligomerization, and Apoptotic Cell Death

Resveratrol has dual effects promoting cell defense, while also having pro-apoptotic effects—in this study we tested its pro-apoptotic activity (Figure 1). It is suggested that resveratrol’s effects depend on the cell type; the compound exhibits pro-survival effects in normal cells but induces apoptosis in cancer cells. Here, we tested its effects on neuroblastoma, SH-SY5Y, cervical cancer, HeLa cells, as well as immortalized human embryonic kidney, HEK-293 cells, which are considered non-cancerous, but have been reported to be tumorigenic. Annexin V–FITC and propidium iodide (PI) staining followed by a flow cytometer analysis showed that resveratrol caused apoptotic cell death in a concertation-dependent manner (Figure 1A–C). SH-SY5Y and HeLa cells were more sensitive to it than HEK-293, with cancer cells showing higher mortality levels compared to HEK-293 cells (Figure 1D).

In addition, cell survival was assayed using XTT, which showed that resveratrol decreased the viability of SH-SY5Y cells in a concentration-dependent manner (Figure 1E).

Both our study and others have demonstrated that the induction of apoptosis leads to VDAC1 overexpression and oligomerization, ultimately resulting in apoptotic cell death [35,36,37,38,39,41,42,44]. SH-SY5Y cells were incubated with resveratrol (75–200 µM), and VDAC1 levels were analyzed by immunoblotting using anti-VDAC1 antibodies (Figure 2A,B). Results show an 8-fold increase in VDAC1 expression levels (Figure 2A,B). Similar results were obtained when VDAC1 expression was analyzed using immunofluorescence (Figure 2C,D), confirming that resveratrol enhances VDAC1 expression.

Next, we investigated the effect of resveratrol on VDAC1 oligomerization using chemical cross-linking with the cell-permeable cross-linker EGS (ethylene glycol bis(succinimidylsuccinate)) and an immunoblotting analysis in SH-SY5Y and HeLa cells. The results clearly demonstrate that resveratrol enhanced VDAC1 oligomerization in a concentration-dependent manner in both SH-SY5Y cells (Figure 2E,F) and in HeLa cells (Figure 2G,H). As previously shown [35], a monomeric VDAC1 form with higher mobility was obtained in the cross-linked samples (Figure 2E,G, arrow), representing the intramolecular cross-linking of monomeric VDAC1 by EGS.

The relationship between VDAC1 overexpression levels, VDAC1 oligomerization, and apoptosis induction was analyzed (Figure 2I,J). We found that these parameters were closely correlated at the resveratrol concentration used, indicating a clear link between VDAC1 overexpression, VDAC1 oligomerization, and apoptosis.

Our findings suggest that, at the concentrations tested, resveratrol enhances VDAC1 expression levels, leading to its oligomerization and the formation of a large channel. This mega-channel facilitates the release of pro-apoptotic proteins, ultimately triggering apoptosis.

### 2.2. Resveratrol Induces Elevation of Intracellular [Ca^2+^] Levels and Increased ROS Production

We evaluated the effects of resveratrol on several parameters associated with apoptosis. Apoptosis induction typically disrupts cellular Ca^2+^ homeostasis, leading to elevated intracellular Ca^2+^ [Ca^2+^]i levels [44]. Its effect on [Ca^2+^]i was monitored using Fluo-4, a fluorescence Ca^2+^ indicator, and a FACS analysis (Figure 3A,B). The results clearly demonstrate that resveratrol significantly increased cellular Ca^2+^ levels.

In addition, it has been shown that many apoptosis inducers lead to an increase in mitochondrial ROS levels. Thus, we tested resveratrol’s ability to increase ROS production using MitoSOX™ Red, a mitochondrial superoxide indicator, and found that it elevated these ROS levels (Figure 3C,D). Furthermore, resveratrol also increased cytosolic ROS levels, as indicated by a DCF analysis (Figure 3E). These findings are consistent with previous reports showing that resveratrol induces cytosolic ROS generation [45,46,47].

Thus, as other apoptosis inducers, resveratrol was found to elevate intracellular Ca^2+^ levels, as well as increase ROS production levels.

Resveratrol is considered to be a major activator of SIRT1, with numerous studies suggesting that its effects are mediated through the activation of SIRT1 [48]. Therefore, we examined whether resveratrol, under the conditions used to activate cell death, increases SIRT1 expression levels. Using immunoblotting with anti-SIRT1 antibodies, we confirmed that it indeed enhances these levels (Figure 3F,G). However, upon cell treatment with resveratrol, no linear correlation was found between the increase in VDAC1 and SIRT1 expression levels.

### 2.3. Resveratrol Induces HK Detachment

Next, we analyzed the effect of resveratrol on mitochondrion-bound HK. It is known that HK bound to VDAC1 prevents the release of pro-apoptotic factors, and that its detachment from the mitochondria promotes cell death [41,42]. Therefore, we examined whether resveratrol also induces HK-I detachment (Figure 4).

To assess the localization of cytosolic and mitochondrion-bound HK-I, cells were treated with resveratrol and stained with anti-HK-I antibodies for immunofluorescence. In untreated cells, HK-I staining was punctuated, suggesting that it bound to the mitochondria. In contrast, resveratrol-treated cells showed diffuse HK-I staining throughout the cell, indicating its detachment from the mitochondria/VDAC1 (Figure 4A). Quantitative analysis of HK-I staining showed a significant increase in its intensity in resveratrol-treated cells (Figure 4B). This increase may indicate a rise in its expression level. However, it could also result from the fluorescence intensity being influenced by HK-I’s concentration at its binding sites in the mitochondria, as opposed to being free in the cytosol.

To further confirm the detachment of HK from the mitochondria, following resveratrol treatment, we analyzed the presence of HK in the cytosol. Cells were permeabilized with digitonin, and the cytosolic and mitochondrial fractions were separated by centrifugation. Cytosolic HK-I was detected in the supernatant, while mitochondrion-bound HK-I was found in the pellet. Resveratrol increased the level of cytosolic HK-I in a concentration-dependent manner (Figure 4C).

These findings suggest that resveratrol, by binding to HK-I or VDAC1 induces conformational changes in HK-I or VDAC1, leading to HK-I detachment from VDAC1, promoting apoptosis.

### 2.4. Resveratrol-Induced Apoptosis Is Reduced in Cells with VDAC1 Depletion

To determine if resveratrol-induced apoptosis is associated with increased VDAC1 expression levels, we silenced VDAC1 expression using specific si-RNA (si-m/h-VDAC1-B) (50 nM and 75 nM) using jetPrime transfection reagent. VDAC1 expression levels were analyzed by immunoblotting and quantitative analysis (Figure 5A,B).

To assess the resveratrol-induced cell death on cells with low VDAC1 expression, we analyzed VDAC1 expression and apoptosis with PI staining and flow cytometry (Figure 5C). The results demonstrate that resveratrol’s ability to induce apoptosis was significantly diminished in the cells with reduced VDAC1 levels, depending on the resveratrol concentration, with apoptosis decreasing by approximately 65% at a concentration of 200 μM (Figure 5C).

These results support the suggestion that resveratrol-induced apoptosis is mediated through VDAC1.

Resveratrol has several analogs, some of which share similar effects, while others offer enhanced benefits. One of these is TSG (trans-2,3,5,4′-tetrahydroxystilbene-2-O-glucoside) (Figure 6A,B). Similar to resveratrol, TSG is known to exert strong anti-inflammatory, antioxidative, and anti-apoptotic effects, in addition to its ability to scavenge free radicals. However, unlike resveratrol, TSG is more hydrophilic, which could influence its cell permeability and bioavailability [49,50].

We compared cell death induction by resveratrol and TSG in SH-SY5Y cells (Figure 6C). The results show that, at similar concentrations, resveratrol induced apoptosis, whereas TSG did not. We also tested the effect of TSG on cell survival and found that, in contrast to resveratrol, which caused a significant reduction in the percentage of live cells, TSG had no effect on cell survival (Figure 6D).

Next, we evaluated the effect of TSG on VDAC1 expression levels (Figure 6E,F) and its oligomerization (Figure 6G,H). As previously shown, resveratrol induced both VDAC1 overexpression and its oligomerization, whereas TSG had no effect on either. These findings reinforce the close association between VDAC1 overexpression, oligomerization, and apoptosis induction.

**Figure 6 ijms-26-03963-f006:**
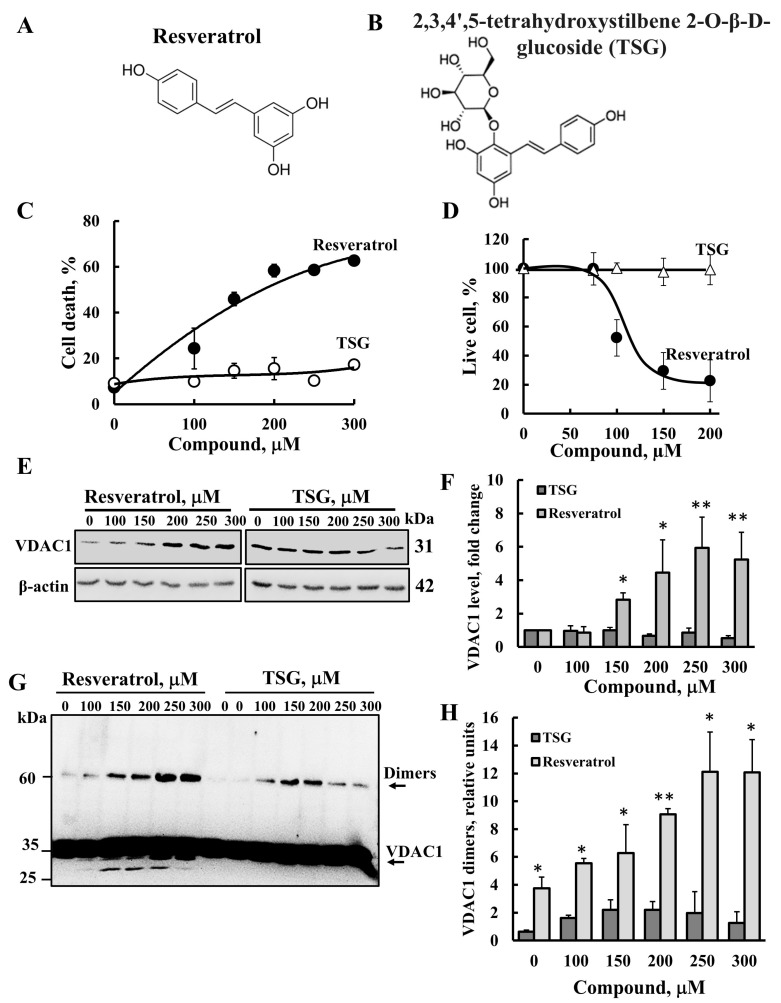
**Resveratrol and TSG directly bind to VDAC1**, **but only resveratrol induced apoptosis.** Chemical structures of resveratrol (**A**) and its analog trans-2,3,5,4′-tetrahydroxystilbene-2-O-glucoside (TSG) (**B**). SH-SY5Y cells were incubated for 24 h with the indicated concentrations of resveratrol or TSG and then analyzed for cell death using PI staining and flow cytometry (**C**), and for cell viability, XTT was used (**D**). Cells were also analyzed for VDAC1 expression levels (**E**) by immunoblotting using anti-VDAC1-specific antibodies. Immunoblotting with β-actin as a loading control is also shown. The levels of VDAC1 were quantified (**F**). In addition, cells were analyzed for VDAC1 oligomerization by incubation (1 mg/mL protein) with the cross-linking reagent EGS (100 μM; 15 min), followed by immunoblotting using anti-VDAC1 antibodies. The positions of the VDAC1 monomers and dimers are indicated (**G**) the red arrow points to monomeric VDAC1 undergoing intramolecular cross-linking. The level of the VDAC1 dimers was analyzed using ImageJ software (version 1.54p) (**H**) and is presented relative to its levels in control cells. Results represent the means ± SEM (n = 3); *p* < 0.05 (*), *p* < 0.01 (**).

Moreover, we tested the ability of resveratrol and TSG to directly bind with purified VDAC1 (Figure 7A,B) using a microscale thermophoresis (MST) interaction assay. The binding of resveratrol and TSG to fluorescently labeled VDAC1 was monitored as a function of time, measuring the fluorescence depletion in a heated spot as a function of the concentration of the interacting partner. Resveratrol binding to VDAC1 as a function of its concentration is shown as thermal migration (Figure 7C). The results indicate that both compounds bound to VDAC1 with comparable affinities, with 50% of maximal binding (C50) observed at 75 μM for both resveratrol and TSG (Figure 7C).

Although both resveratrol and TSG similarly bound to VDAC1, only resveratrol induced VDAC1 overexpression, which led to its oligomerization and subsequent cell death.

## 3. Discussion

Resveratrol exhibits dual effects, supporting cell defense and extending cell lifespan in various tissues under stress, while also inducing apoptosis. The outcome of its action is influenced by several factors. (a) Cell type: in normal cells, resveratrol enhances cell function and longevity. However, in cancer cells, which are often resistant to cell death, it triggers apoptosis. For instance, in an experiment with cardiomyocytes, resveratrol induced cell death in cancerous cells, but not in normal cells from the same tissue [15]. (b) Concentration-dependent effects: at low concentrations, resveratrol acts protectively, improving cell longevity and reducing ROS levels due to its antioxidant properties. Conversely, at high concentrations, it increases ROS levels, producing pro-oxidant effects that lead to apoptosis [51].

### 3.1. Resveratrol, Its Analog TSG, Mitochondria, VDAC1, and Cell Death

In this study, we investigated whether resveratrol, like many apoptosis inducers [37] triggers apoptotic cell death through VDAC1, and characterized the underlying mechanism of its VDAC1-mediated activity. The involvement of VDAC1 in resveratrol’s pro-apoptotic activity was demonstrated through the following several approaches: (A) resveratrol significantly increased VDAC1 expression levels (Figure 2A–D and Figure 3F); (B) it induced VDAC1 oligomerization (Figure 2E–H and Figure 6G,H); (C) the reduction in resveratrol’s pro-apoptotic activity in cells with VDAC1 was silenced via specific si-RNA; and (D) by the direct interaction of resveratrol with VDAC1, as shown by the MST method (Figure 7). This direct interaction suggests that resveratrol may modulate VDAC1 conformation, leading to its oligomerization.

However, the finding that the resveratrol analog TSG, despite binding to VDAC1 with similar affinity (Figure 7), did not induce VDAC1 oligomerization or cell death (Figure 6), indicates that VDAC1 overexpression is a key factor in resveratrol’s ability to induce cell death.

Resveratrol’s ability to increase VDAC1 expression aligns with previous reports indicating that it modulates gene expression [25,52]. It downregulates the expression of anti-apoptotic proteins such as Bcl-2, Bcl-xL, survivin, and XIAP, while upregulating pro-apoptotic proteins like Bax, Bak, PUMA, Noxa, P21, Bim, various caspases, and other pro-apoptotic factors [21,22]. Additionally, resveratrol modulates the expression of tumor suppressor genes, such as AMPK and p53, which are linked to apoptosis and the prevention of tumorigenesis [23,30,31].

The precise mechanism by which resveratrol regulates gene expression remains unclear. It is suggested that it mediates the activity of several transcription factors, including activator protein 1 (AP-1), NF-κB, β-catenin [53,54,55], FOXM1, STAT3, HIF-1α, NRF2 [56], Egr-1, Elk-1, and Nrf2 [57]. Furthermore, resveratrol is considered a major activator of SIRT1, which is involved in the deacetylation of various transcription factors [15,58,59]. Thus, through its influence on these transcription factors, resveratrol regulates a wide array of gene expressions, including VDAC1. By influencing different transcription factors, it is involved in regulating various gene expressions, including that of VDAC1.

In this study, we demonstrated that resveratrol increases intracellular Ca^2+^ levels ([Ca^2^+]i) (Figure 3A,B) and activates signaling pathways that regulate gene transcription [60]. Our findings show that various apoptosis inducers elevate cytosolic Ca^2+^, leading to VDAC1 overexpression, oligomerization, and apoptosis [44,61]. Conversely, reducing [Ca^2+^]i with the cell-permeable Ca^2+^-chelating agent BAPTA-AM inhibited VDAC1 overexpression and prevented apoptosis [44,61].

A variety of regulatory mechanisms involved in Ca^2+^-dependent gene expression have been reported [60,62,63]. These mechanisms include the regulation of mRNA transcription, elongation, splicing, stability, and translation [60]. Additionally, Ca^2+^-responsive elements (CaRE) or Ca^2+^-dependent transcription factors, activated by calcineurin, are known to trigger the expression of hypertrophic response genes [64,65,66]. Therefore, resveratrol either directly or via Ca^2+^ activates transcription factors that trigger the overexpression of VDAC1, promoting its oligomerization, and ultimately, leading to apoptosis.

Resveratrol has a number of analogs with varying properties due to structural differences, such as additional hydrophilic groups [7,49,50,67,68]. One such analog, TSG, has been reported to primarily offer cell protection, unlike resveratrol, which also induces apoptosis [49,50]. Our study showed that TSG bound to VDAC1 with the same affinity as resveratrol (Figure 7), but in contrast to resveratrol, did not induce apoptosis, even at high concentrations (Figure 6). Moreover, TSG did not promote VDAC1 overexpression, suggesting that it does not influence gene transcription.

These findings further emphasize the connection between VDAC1 overexpression and the induction of cell death. Consequently, we propose VDAC1 as a new target for resveratrol, revealing a novel mechanism by which resveratrol induces cell death (Figure 8). In this content, VDAC1 overexpression has been implicated in many diseases including cancer, Alzheimer’s disease (AD) [69,70,71,72], Parkinson’s disease (PD), amyotrophic lateral sclerosis (ALS) [73,74], type 2 diabetes (T2D) [75,76,77], cardiovascular diseases [78,79], and autoimmune diseases [80]. This aligns with the observation that VDAC1 overexpression is linked to disease progression, whereas lowering VDAC1 levels can alleviate the disease.

A recent study [81] demonstrated that VDAC1 is overexpressed in PD-like pathology, such as in A53T transgenic mice or cells exposed to MPP+ (1-Methyl-4-phenylpyridine), both of which induce PD-like conditions. At both conditions, resveratrol, at low concentrations (10–40 µM), decreased the overexpression of VDAC1, reduced mtDNA release (which is mediated by VDAC1 oligomers [80]), inhibited mPTP opening, and prevented mitochondrial dysfunction. These effects helped mitigate dopaminergic neuron degeneration and pathological progression in a Parkinson’s disease model [81]. These findings support the involvement of VDAC1 in the protective effects of resveratrol at low concentrations, including its antioxidant, anti-inflammatory, and neuroprotective properties, as well as improvements in cardio-metabolic health and anti-aging effects [14,15,16,17,18,19]; furthermore, at high concentrations, it promotes cell death and acts as an anti-cancer agent [9,10,11,12].

### 3.2. Resveratrol-Induced Cell Death: Proposed Mode of Action

It is proposed that resveratrol’s primary chemotherapeutic effect is apoptosis, which is linked to the activation of the tumor suppressor protein p53 and the death receptor Fas/CD85/APO-1 across various cancer-cell types [82].

As illustrated in Figure 8, we present a new mechanism for resveratrol’s induction of cell death, which involves the overexpression of VDAC1 and its subsequent oligomerization. Previous studies have demonstrated that apoptosis inducers, although proposed to elicit cell death via different mechanisms, all promote VDAC1 overexpression and oligomerization [35,36,38,39,41,44,61]. When VDAC1 expression levels reach high levels, its monomers form oligomers, creating a large channel, allowing the release of apoptotic factors including Cyto *c.* The Cyto *c* participates in apoptosome formation that results in apoptosis activation (Figure 8) [35,36,38,39,41,42,44]. This suggests that VDAC1 overexpression represents a common pathway leading to apoptosis.

Several additional mechanisms have been proposed for resveratrol’s action that may also be related to VDAC1. Among these is the activation or inhibition of SIRT1, which can vary depending on whether resveratrol is promoting cell death or conferring cell protection [15,58,59]. SIRT1 plays a key role in cell proliferation, differentiation, and the prevention of apoptotic death [83]. However, its role in oncogenesis is debated, as it can act as a tumor suppressor or an oncogene, or conversely, it may have no impact on cancer progression [84]. In this study, we observed that resveratrol also increased SIRT1 expression levels in parallel with VDAC1 overexpression (Figure 3F,G). Since SIRT1 is a deacetylase, it may remove acetyl groups from VDAC1, thereby influencing its conformation and possibly its oligomeric state.

We showed that resveratrol detaches HK-I from VDAC1 (Figure 4). Cancer cells overexpress mitochondria-bound HK-I and HK-II to insure an energy supply and protection against mitochondria-mediated apoptosis [41,42,85,86,87]. The association of HK with VDAC1 offers several advantages to cancer cells, such as direct access to mitochondrial ATP, thus, coupling cytosolic glycolysis to mitochondrial oxidative phosphorylation. This interaction regulates cellular metabolism by controlling the glycolytic pathway, which generates metabolic intermediates that are essential for cancer-cell survival. Additionally, the binding of HK to VDAC1 also protects against mitochondria-mediated apoptosis [41,42,85,86], reducing intracellular ROS levels, and enhancing cholesterol synthesis and uptake [36].

Given that HK binding to VDAC1 provides both metabolic benefits and apoptotic protection, it has become an important target for anti-cancer therapies [88]. To disrupt the HK–VDAC1 association, various compounds have been developed, including peptides derived from HK-I [89] and methyl-jasmonate [90], and VDAC1-based peptides [41,85].

In this study, we demonstrated that resveratrol induces the detachment of HK from VDAC1, which both inhibits cancer-cell metabolism and triggers apoptosis.

Furthermore, resveratrol has been shown to suppress HK-II expression and glycolysis in non-small cell lung cancer (NSCLC), with the Akt signaling pathway proposed as a key molecular mechanism underlying its anti-tumor effects [91,92]. Moreover, decreased expression of HK-II has been linked to apoptosis induction, which is expected, as overexpression of HK-I or HK-II typically inhibits apoptosis [93]. Resveratrol has also been shown to reduce glucose uptake and glycolysis by modulating factors such as Glut1, PFK1, HIF-1α, ROS, PDH, and the CaMK/AMPK pathway [94].

Taken together, the VDAC1-associated mechanisms identified here provide insight into how resveratrol can exert such diverse effects—from promoting cell survival to inducing cell death—owing to the multi-functional roles of VDAC1.

### 3.3. Resveratrol as Potential Cancer Therapy

Chemotherapeutic agents like cisplatin, etoposide, and prednisolone are commonly used to treat cancer by inducing cancer-cell death. However, these treatments often have significant side effects, as they do not selectively target cancer cells and can harm normal, healthy cells. A promising strategy to address this issue is to combine traditional chemotherapies with natural compounds such as resveratrol, a Stilbene family molecule found in various plants [53,95]. Resveratrol has emerged as a potential anti-cancer agent because it induces a cytotoxic effect on cancer cells, while sparing non-malignant cells [15,96,97,98,99,100].

Numerous studies have explored the anti-cancer effects of resveratrol and investigated its underlying cellular mechanisms and signaling pathways both in vitro and in vivo [26,101,102]. Resveratrol can be considered as a natural chemotherapy agent that has potentially fewer side effects. In-vitro studies have shown that resveratrol affects various types of cancer across different stages, from initiation and promotion to progression, by activating a wide array of signaling pathways that are involved in controlling cell growth, division, metastasis, inflammation, cell death, and angiogenesis [24,103,104]. Its anti-cancer activity is also linked to its pro-apoptotic properties [51,105,106,107]. Additionally, resveratrol has anti-inflammatory effects that reduce the risk of cancer, as chronic inflammation is known to contribute to cancer development [54,55,108].

In our study, we demonstrated that relatively high concentrations of resveratrol induce VDAC1-dependent apoptosis and activate various apoptosis-associated processes that include increases in intracellular Ca^2+^ levels, ROS production, and HK detachment, as summarized in Figure 8.

In conclusion, this study proposes that resveratrol’s pro-apoptotic activity is driven by VDAC1 overexpression, which promotes its oligomerization, ultimately, triggering apoptotic cell death. In addition, resveratrol activates other processes associated with apoptosis such as elevated cytosolic Ca^2+^ and ROS production, and the detachment of mitochondria-bound HK. The detachment of HK from VDAC1 not only disrupts energy production but also facilitates VDAC1 oligomerization and apoptosis. Thus, resveratrol, with its natural origin and fewer side effects, represents a promising candidate for cancer therapy.

## 4. Materials and Methods

### 4.1. Materials

Resveratrol, bovine serum albumin (BSA), trypan blue, Triton X-100, Tween-20, 4′,6-diamidino-2-phenylindole (DAPI), dimethyl sulfoxide (DMSO), tetrahydroxystilbene 2-O-β-D-glucoside-5,′2,3,4 (TSG), dl-dithiothreitol (DTT), propidium iodide (PI), HEPES, leupeptin, phenylmethylsulfonyl fluoride (PMSF), and tris were obtained from Sigma-Aldrich (St. Louis, MO, USA). Dulbecco’s modified Eagle’s medium (DMEM), normal goat serum (NGS), fetal bovine serum (FBS), trypsin, and the supplement penicillin-streptomycin were obtained from Gibco (Grand Island, NY, USA). Hank’s Balanced Salt Solution (HBSS) and an XTT-based cell survival kit was purchased from Biological Industries (Beit Ha-Emek, Israel). EGS was obtained from Pierce (Appleton, WI, USA). Annexin V-fluorescein isothiocyanate (FITC) was from Enzo Life Sciences (Lausanne, Switzerland). Protease inhibitor cocktail set III and digitonin was obtained from Calbiochem (Nottingham, UK). Fluo-4-AM, MitoSOX Red and dichlorofluorescein (DCF) were acquired from Thermo Fisher Scientific (Waltham, MA, USA). JetPrime cell transfection reagent was obtained from Polyplus (IIIkrich, France). si-m/h-VDAC1-B and si-NT were synthesized and purchased from GenePharma (Suzhou, China). Fluoroshield was obtained from Immuno Bio Science Corporation (Washington, DC, USA).

Primary and secondary antibodies, their source and the dilutions used are detailed in Table 1.

### 4.2. Cell Culture and Treatment with Resveratrol and TSG

SH-SY5Y (human neuroblastoma), HeLa (human cervix adenocarcinoma), and HEK-293 (human epithelial kidney) cell lines were obtained from ATCC and cultured in DMEM, supplemented with 10% FBS, 100 U/mL penicillin, and 100 μg/mL streptomycin. The cells were maintained in a humid atmosphere at 37 °C and 5% CO_2_.

For treatment with resveratrol or TSG, cells at approximately 80% confluence were incubated for 24 h with various concentrations of resveratrol or TSG, both dissolved in DMSO and subsequently diluted in the cell culture medium to the desired concentrations. The final DMSO concentration in both control and treated samples was 0.28%. After treatment, the cells were harvested, centrifuged (1500× *g* for 5 min), washed with PBS, and then subjected to the intended assays.

### 4.3. Protein Extraction, Gel Electrophoresis, and Immunoblotting

Cells subjected to the desired treatment were lysed using lysis buffer (50 mM Tris-HCl, pH 7.5, 150 mM NaCl, 1 mM EDTA, 1.5 mM MgCl_2_, 10% glycerol, 1% Triton X-100), and supplemented with a protease inhibitor cocktail (Calbiochem; San Diego, CA, USA). The lysates were vortexed and incubated for 15 min on ice. After incubation, the lysates were centrifuged at 15,000× *g* for 10 min at 4 °C, and the protein concentration of the supernatant was measured. Protein samples were stored at −80 °C until further use in gel electrophoresis. Protein aliquots (10–20 μg) were subjected to SDS-PAGE and then were electro-transferred onto nitrocellulose membranes for immunostaining. The membranes were first blocked by incubation (2 h) with a solution containing 5% non-fat dry milk and 0.1% Tween-20 in tris-buffered saline (TBST, pH 7.8), followed by incubation with primary antibodies (as listed in Table 1). The membranes were then incubated with HRP-conjugated anti-mouse or anti-rabbit IgG as secondary antibodies. Enhanced chemiluminescent substrate EZ-ECL (Biological Industries; Beit Ha-Emek, Israel) was used to visualize HRP activity. Band intensities were analyzed using FUSION-FX (Vilber Lourmat, France, version V18.02) or ImageJ software version 1.54p (Bethesda, MD, USA).

### 4.4. Cross-Linking Experiments

Cells were grown in 6-well plates (60% confluence) subjected to resveratrol or TSG treatment, harvested, and washed with PBS, and their protein concentration was determined. Cells (3 mg/mL or 1 mg/mL) were incubated with the cross-linking reagent EGS (300 μM or 100 μM) in PBS at pH 8.3 for 15 min. Samples (60–80 µg protein) were then subjected to SDS-PAGE. For visualization of VDAC1 oligomerization, nitrocellulose membranes were treated with 0.1 mM glycine at pH 2.2 prior to immunoblotting and washed several times with 0.1% Tween-20 in Tris-buffered saline (TBST). Enhanced chemiluminescent substrate EZ-ECL (Biological Industries; Beit Ha-Emek, Israel) was used to detect HRP activity. Quantitative analysis of immunoreactive VDAC1 dimers was performed using ImageJ software (Bethesda, MD, USA, version 1.54p).

### 4.5. Apoptosis and Cell Death Analyses

Cell death was analyzed by propidium iodide (PI) staining (final concentration of 6.25 μg/mL), followed by flow cytometry with an iCyt sy3200 Benchtop Cell Sorter/Analyzer (Sony Biotechnology Inc.; San Jose, CA, USA) and analysis with EC800 software. Apoptosis was analyzed by PI and Annexin V-FITC staining, carried out according to the manufacturer’s instructions. After treatment, cells were harvested (1500 g, 5 min), washed and re-suspended in 200 µL of binding buffer (10 mM Hepes/NaOH, pH 7.4, 140 mM NaCl, and 2.5 mM CaCl_2_). Annexin V-FITC/PI staining was performed, and the samples were analyzed by flow cytometry. At least 10,000 events were recorded, represented as dot plots.

### 4.6. Cell Viability, XTT Assay

Cells (at 60% confluence) were analyzed for cell survival using an XTT assay, according to the manufacturer’s instructions. In this essay, mitochondrial dehydrogenases in live cells convert the XTT solution into an orange formazan product. Briefly, the activation solution was added to the XTT solution, which was then applied to the cells, followed by incubation for 1–2 h in the dark (5% CO_2_, 37 °C). Absorbance at 450–500 nm was analyzed using an Infinite M1000 plate reader (Tecan; Männedorf, Switzerland). Cell viability was displayed as the percentage of treated cells relative to the control.

### 4.7. Intracellular Ca^2+^ Level Analysis

Fluo-4-AM was used to monitor changes in cytosolic Ca^2+^ levels. Cells were subjected to the desired treatment, then harvested, collected (1500× *g* for 5 min), washed with HBSS supplemented with 1.8 mM CaCl_2_ (HBSS+), and incubated with 2 μM Fluo-4 in 200 μl HBSS(+) for 30 min at 37 °C in a light-protected environment. After removing the excess dye by washing with HBSS(+), the cellular free Ca^2+^ concentration was promptly measured using an iCyt sy3200 Benchtop Cell Sorter/Analyzer (Sony Biotechnology Inc.; San Jose, CA). At least 10,000 events were recorded by the FL1 detector, represented as a histogram, and analyzed by EC800 software, version 1.3.7 (Sony Biotechnology Inc.). Positive cells showed a shift to an enhanced level of green fluorescence (FL1).

### 4.8. Reactive Oxygen Species (ROS) Level Analysis

To assess mitochondrial or cytosolic ROS levels, following the desired treatment, cells were collected and then treated with MitoSOX Red (5 μM, 10 min at 37 °C) or with DCF (4 μM; 30 min at 37 °C), monitoring mitochondrial and cytosolic ROS, respectively. At least 10,000 events were recorded by the FL2 detector, represented as a histogram, and analyzed with EC800 software (Sony Biotechnology Inc.).

### 4.9. Microscale Thermophoresis (MST) Measurements

An MST analysis was performed using a NanoTemper Monolith NT.115 apparatus, as described previously [109]. Briefly, purified VDAC1 was fluorescently labeled using a NanoTempers Protein-labeling kit BLUE (L001; NanoTemper Technologies; Munich, Germany). A constant concentration of fluorescently labeled VDAC1 (162 nM) was incubated with different concentrations of resveratrol or TSG in MST binding buffer (10 mM Tricine-HCl, 100 mM NaCl, pH 7.4) for 30 min at 37 °C in the dark. Afterwards, 3–5 µL of the samples were loaded into a glass capillary (Monolith NT Capillaries; NanoTemper Technologies; Munich, Germany) and the thermophoresis analysis was performed. The results are presented as the percentage of changes in normalized fluorescence (ΔF Norm).

### 4.10. HK-I Detachment from the Mitochondria

Cells treated with increased concentrations of resveratrol (75–200 µM) for 24 h were harvested, washed twice with PBS at pH 7.4, gently re-suspended at 6 mg/mL in ice-cold buffer (100 mM KCl, 2.5 mM MgCl_2_, 250 mM sucrose, 20 mM HEPES/KOH, pH 7.5, 0.2 mM EDTA, 1 mM dithiothreitol, 1 μg/mL leupeptin, 5 mg/mL cytochalasin B, and 0.1 mM PMSF) containing 0.005% digitonin, and incubated for exactly 10 min on ice. Samples were centrifuged at 10,000× *g* at 4 °C for 10 min to obtain supernatants (cytosolic fraction) and pellets (containing mitochondria). HK-I in the cytosolic fraction was analyzed by immunoblotting using specific anti-HK-I antibodies.

### 4.11. Immunofluorescence (IF) Staining

SH-SY5Y cells were grown on 13-mm glass cover slips in a 12-well plate, treated with resveratrol for 24 h, washed with PBS, and fixed with 4% paraformaldehyde at room temperature for 15 min. Cells were permeabilized by 0.1% Triton-X100 in PBS, pH 7.4 (PBST), and non-specific antibody binding was reduced by incubating the sections for 2 h with 10% normal goat serum (NGS), 1% BSA, and 0.1% Triton-X100 in PBS. Cells were incubated overnight at 4 °C with the primary antibodies (sources and dilutions used are detailed in Table 1) in 5% NGS and 1% BSA in PBS. For IF, fluorophore-conjugated anti-rabbit or anti-mouse as secondary antibodies were used (sources and dilutions used are detailed in Table 1). Nuclei were stained with DAPI (0.07 μg/mL) for 15 min in the dark, and cover slips were carefully washed, dried, and mounted on slides with Fluoroshield mounting medium (Immuno Bio Science Corporation, Washington, DC, USA). After overnight drying at 4 °C, images were acquired using a confocal microscope (Olympus 1X81; Tokyo, Japan).

### 4.12. Cells VDAC1 Silencing

2′-O-methyl-modified human and mouse VDAC1 specific siRNA (si-m/h-VDAC1-B), and non-targeting siRNA (si-NT) were synthesized and obtained from GenePharma (Suzhou, China). The underlined nucleotides were 2′-O-methyl-modified.

si-m/h-VDAC1-B, Sense: 5’-GAAUAGCAGCCAAGUAUCAGtt-3’

Anti-sense: 5’-CUGAUACUUGGCUGCUAUUCtt-3’

si-NT, Sense: 5’-GCAAACAUCCCAGAGGUAU-3’

Anti-sense: 5’-AUACCUCUGGGAUGUUUGC-3’

Cells were seeded (40–60% confluence) in 6-well plates and transfected with 50 nM or 75 nM si-m/h-VDAC1-B or si-NT using JetPrime transfection reagent (Polyplus; IIIkrich, France) for 24 h, according to the to the manufacturers’ protocol. Post transfection, cells were subjected to treatment with resveratrol (24 h) and then were subjected to immunoblotting for VDAC1 expression levels using anti-VDAC1 specific antibodies.

### 4.13. Statistical Analysis

Data are presented as the means ± SEM of at least three independent experiments (n = 3), unless otherwise specified. Differences between groups were assessed using a two-tailed Student’s *t*-test, with significance determined using the *t*-test function in Microsoft Excel. Statistical significance is indicated as follows: *p* ≤ 0.05 (*), *p* ≤ 0.01 (**), *p* ≤ 0.001 (***), or *p* ≤ 0.0001 (****).

## Figures and Tables

**Figure 1 ijms-26-03963-f001:**
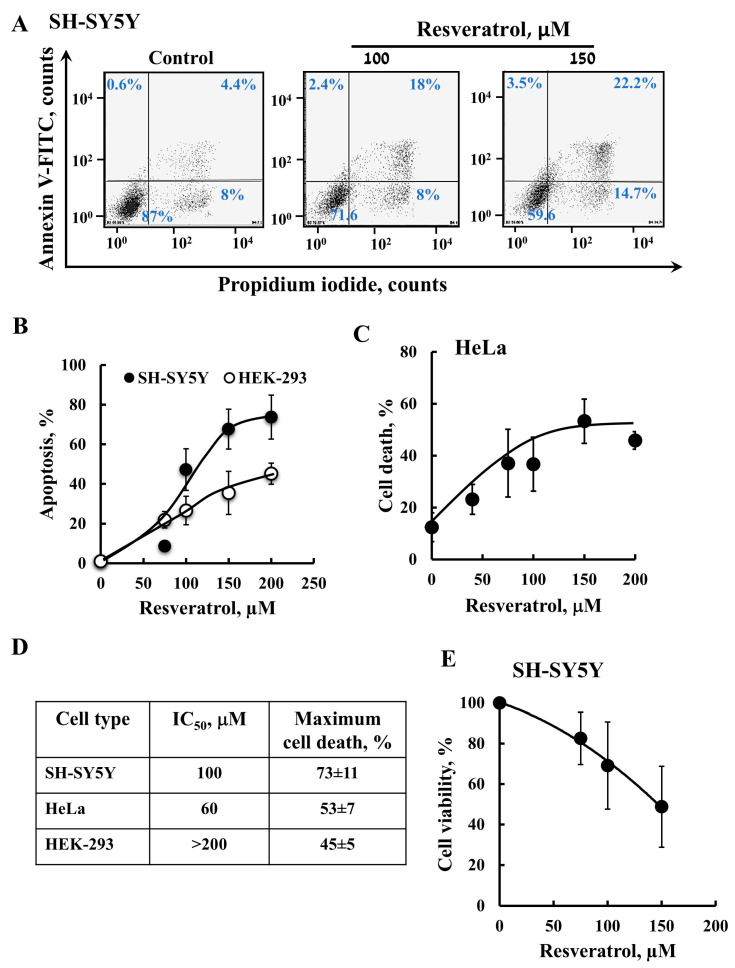
**Resveratrol induces apoptosis and reduces the viability of cancer cells.** SH-SY5Y, HeLa, or HEK-293 cells were treated with the indicated concentrations of resveratrol for 24 h or 48 h. Then, cells were harvested and subjected to an apoptotic cell death analysis (**A**–**D**) or cell viability analysis (**E**). (**A**,**B**) Apoptosis of SH-SY5Y and HEK-293 cells was assayed using annexin V–FITC and PI staining, and flow cytometry and cell death of HeLa cells (**C**) was assayed with PI staining and flow cytometry. Representative histograms of resveratrol induction of apoptosis in SH-SY5Y cells are shown (**A**)**.** Quantitative analyses of three independent experiments as in A are shown (**B**,**C**). Summary of the IC_50_ and maximal cell death obtained for the three cell types (**D**). Cell viability of the SH-SY5Y cells was determined using an XTT-based assay, and viable cells were measured at 490 nm in a microplate reader (**E**). The results are the mean ± SEM (n = 3).

**Figure 2 ijms-26-03963-f002:**
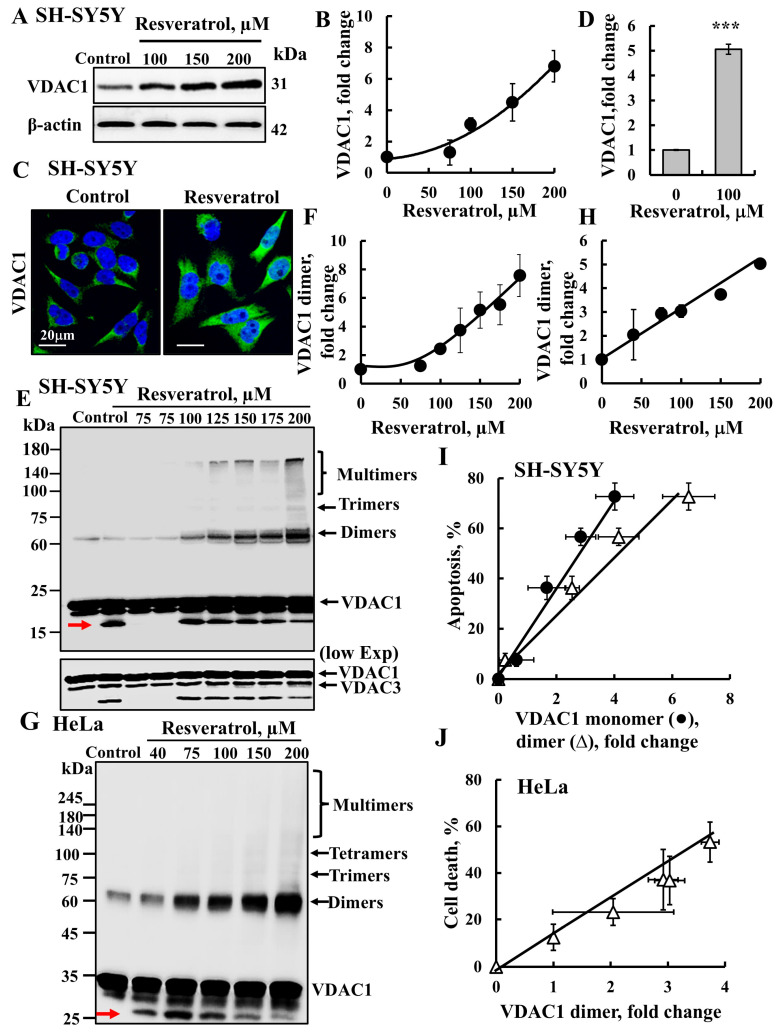
**Resveratrol enhances VDAC1 expression levels and oligomerization**. (**A**,**B**) SH-SY5Y cells were incubated for 24 h with the indicated concentrations of resveratrol and then analyzed for VDAC1 expression level by immunoblotting using anti-VDAC1 antibodies (**A**). Immunoblotting with β-actin as a loading control is also shown. Quantification of VDAC1 levels (**B**). (**C**) SH-SY5Y were seeded on 13-mm glass coverslips, treated with resveratrol (100 μM, 24 h), fixed and subjected to immunofluorescence (IF) using anti-VDAC1 antibodies. (**D**) Quantification of VDAC1 levels of the IF staining. (**E**,**F**) SH-SY5Y cells were incubated for 24 h with the indicated concentrations of resveratrol and analyzed for VDAC1 oligomerization by incubation (3 mg/mL protein) with the cross-linking reagent EGS (300 μM), followed by immunoblotting with anti-VDAC1 antibodies. The positions of the VDAC1 monomers, dimers, trimers, tetramers, and higher oligomers are indicated. The red arrow points to monomeric VDAC1 undergoing intramolecular cross-linking. The level of VDAC1 dimers was analyzed using ImageJ software (version 1.54v, Bethesda, MD, USA) and is presented relative to its levels in the control cells (**F**). (**G**,**H**) HeLa cells were incubated for 48 h with the indicated concentrations of resveratrol and analyzed for VDAC1 oligomerization (1 mg/mL protein), and incubated with EGS (100 μM), followed by immunoblotting (**G**) and dimer quantification (**H**). (**I**,**J**) Correlation plot of apoptosis percentage relative to the increase in VDAC1 monomers (●) and its dimer (Δ) levels obtained at the same concentration of resveratrol for SH-SY5Y (**I**) or HeLa (**J**). Results represent the means ± SEM; *p* < 0.001 (***).

**Figure 3 ijms-26-03963-f003:**
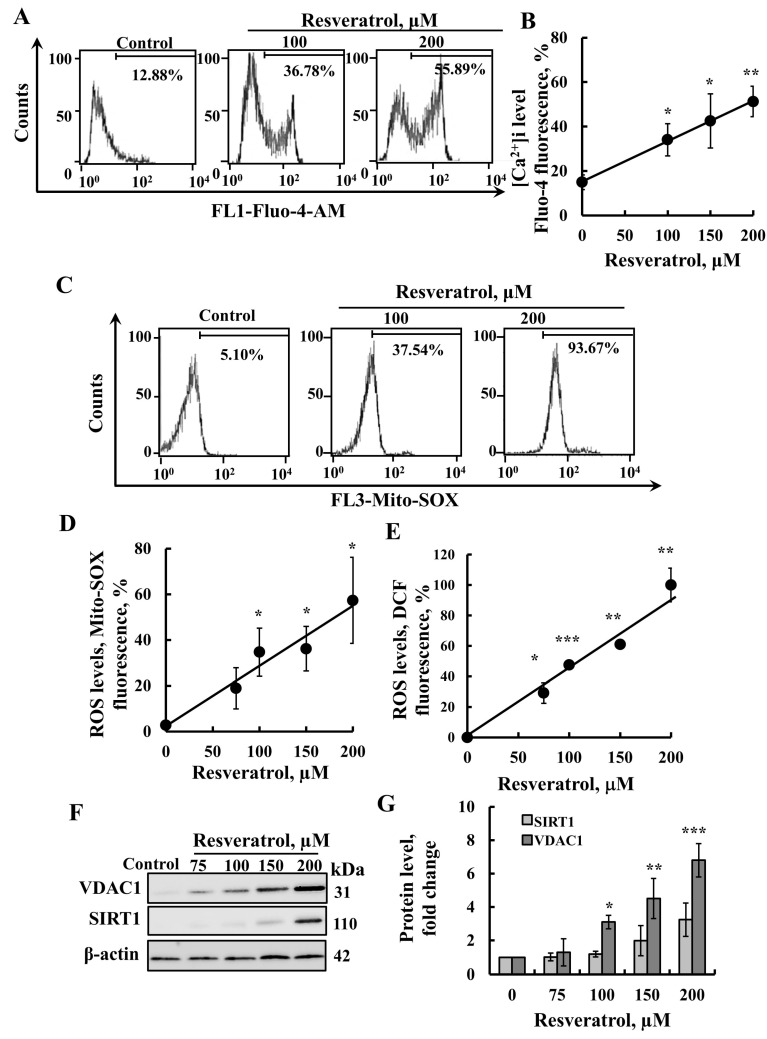
**Resveratrol elevates cellular Ca^2+^ levels and increases ROS production.** SH-SY5Y cells were incubated for 24 h with the indicated concentrations of resveratrol. They were then harvested, and intracellular calcium ([Ca^2+^]i) levels (**A**,**B**), mitochondrial, and cellular superoxide (ROS) levels (**C**–**E**) were measured using Fluo-4-AM, MitoSOX Red, and DCF, respectively, and analyzed by flow cytometry. Representative FACS histograms (**A**,**C**) and quantification (**B**,**D**,**E**) are presented. (**F**) Cells were treated as in A, and then analyzed for VDAC1 or SIRT1 expression levels by immunoblotting using anti-VDAC1 or anti-SIRT1 specific antibodies. Immunoblotting with β-actin as a loading control is also shown. (**G**) Quantification of VDAC1 and SIRT1 levels. Results represent the means ± SEM (n = 3); *p* < 0.05 (*), *p* < 0.01 (**), *p* < 0.001 (***).

**Figure 4 ijms-26-03963-f004:**
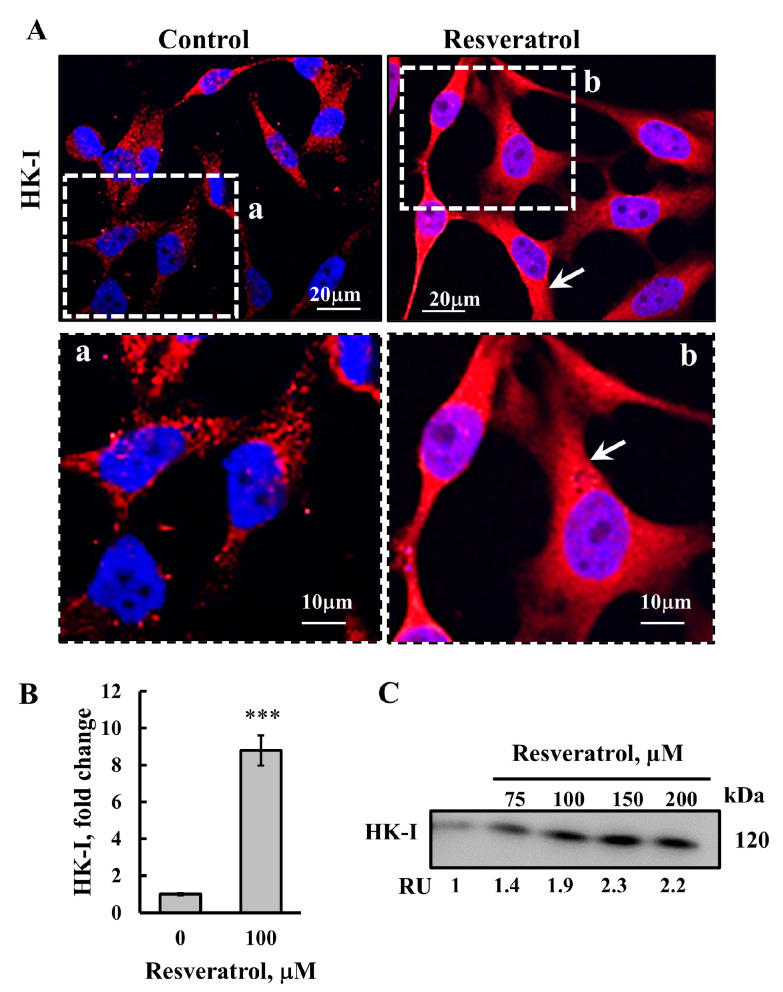
**Resveratrol induces HK-I detachment from the mitochondria.** SH-SY5Y cells were seeded on 13-mm glass coverslips and then incubated for 24 h with resveratrol (100 μM) followed by IF staining with anti-HK-I antibodies. Images were visualized by confocal microscopy (**A**). Enlargements (a, b) of dashed squares are presented. The arrows point to cytosolic, diffused staining of HK-I. (**B**) Quantification of HK-I expression levels. (**C**) SH-SY5Y cells were treated with the indicated concentrations of resveratrol, which were then harvested and incubated on ice for 10 min with 0.005% digitonin. Following centrifugation (10,000× *g*, 5 min), the supernatants (cytosol) were subjected to SDS-PAGE and immunoblotting using anti-HK-I antibodies. Relative HK-I levels released to the cytosol are presented below the immunoblots relative to the control in relative units (RUs). Results represent the means ± SEM (n = 3); *p* < 0.001 (***).

**Figure 5 ijms-26-03963-f005:**
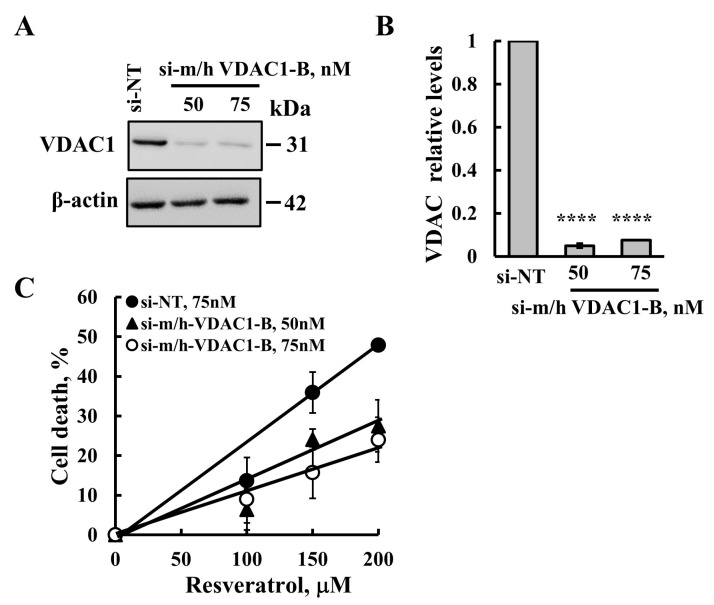
**VDAC1 silencing reduced resveratrol-induced apoptosis.** SH-SY5Y cells were transfected with si-NT (75 nM) or with si-m/h-VDAC1-B (50 nM or 75 nM), using JetPrime transfection reagent, as described in the Section 4. At 24 h post transfection, cells were treated with the indicated concentrations of resveratrol for 24 h and subjected to immunoblotting for VDAC1 expression levels using anti-VDAC1 specific antibodies (**A**). (**B**) VDAC1 expression levels were quantified and are presented as relative units (RUs). (**C**) Cell death induced by resveratrol in SH-SY5Y treated with si-NT (●) or treated with si-m/h-VDAC1-B 50 nM (▲) or 75 nM (○), determined using PI staining and a flow cytometer analysis. Results reflect the mean ± SEM (n = 3), *p* ≤ 0.0001 (****). 2.5. Resveratrol and its Analog Directly Bind to VDAC1, but Only Resveratrol Induces Apoptosis.

**Figure 7 ijms-26-03963-f007:**
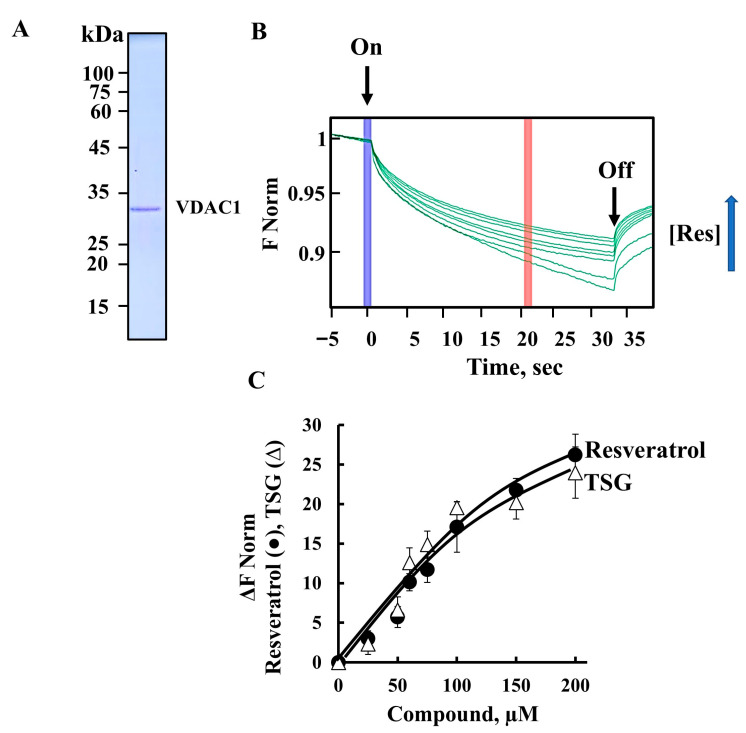
**Resveratrol and TSG directly bind to VDAC1.** (**A**) Coomassie bluestained purified VDAC1 was fluorescently labeled using a NanoTemper Protein-Labeling Kit BLUE. Labeled VDAC1 was incubated for 30 min with the indicated concentrations of resveratrol, and then 3–5 μL of the samples were loaded into MSTgrade glass capillaries, and the thermophoresis process was measured. (**B**) Schematic presentation of the MST assay. Thermal-migration curves of VDAC1 in the presence of different concentrations of resveratrol upon starting heat (On) and stopping heating (Off). (**C**) VDAC1 binding in the presence of resveratrol (●) or TGS (Δ) derived from the MST measurements are presented. Results represent the means ± SEM (n = 3).

**Figure 8 ijms-26-03963-f008:**
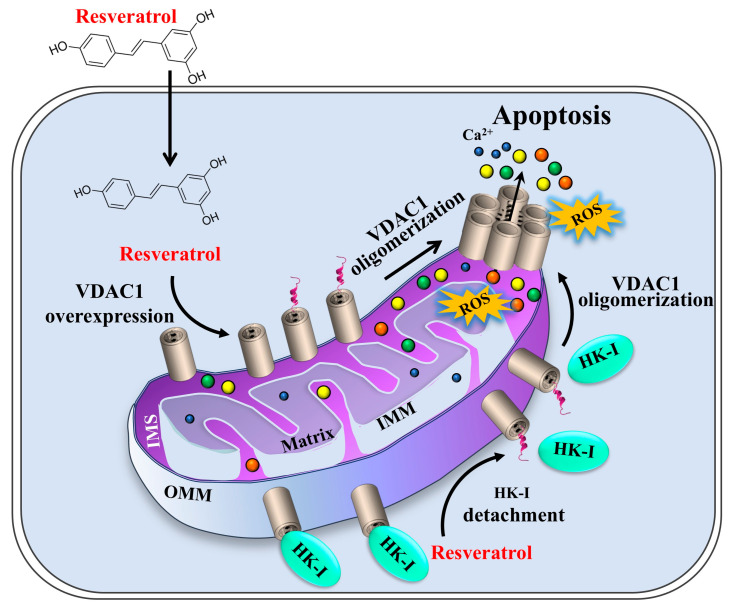
**Proposed model for resveratrol inducing VDAC1 overexpression**, **oligomerization**, **and apoptosis.** Resveratrol entering the cell enhances VDAC1 expression levels with the overexpressed VDAC1, shifting the equilibrium to the VDAC1 oligomeric state. This mediates the release of Ca^2+^, ROS, and apoptogenic proteins, leading to apoptosis. Resveratrol detaches HK-I from VDAC1, further enhancing VDAC1 oligomerization, which results in apoptosis and disrupted metabolism.

**Table 1 ijms-26-03963-t001:** Antibodies used in this study. Antibodies against the indicated protein, their catalog number, source, and the dilutions used in immunoblotting (WB) and immunofluorescence (IF) experiments are listed.

Antibody	Source and Cat. No.	Dilution	
		WB	IF
Rabbit polyclonal anti-VDAC1	Abcam; Cambridge, UK, ab15895	1: 5000	
Mouse monoclonal anti-VDAC1	Abcam; Cambridge, UK, ab186321		1:500
Rabbit monoclonal anti-HK-I	Abcam; Cambridge, UK, ab150423	1:2000	1:500
Rabbit polyclonalanti-SIRT1	Millipore; Billerica, MA, USA, 07-131	1:10,000	
Mouse monoclonal anti-β-actin	Millipore; Billerica, MA, USA, MAB1501	1:40,000	
Donkey anti-mouse (Alexa Fluor 488)	Abcam; Cambridge, UK, ab150109		1:750
Goat anti-rabbit (Alexa Fluor 555)	Abcam; Cambridge, UK, ab150086		1:850
Goat anti-rabbit (HRP)	Promega; Madison, WIS, USA W4018	1:15,000	
Donkey anti-mouse (HRP)	Abcam; Cambridge, UK, ab98799	1:10,000	

## Data Availability

The authors agree that all data and materials generated from this study will be shared with other qualified investigators through academically established means.

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
