# Peer review of "Resveratrol’s Pro-Apoptotic Effects in Cancer Are Mediated Through the Interaction and Oligomerization of the Mitochondrial VDAC1"

_ijms, 2025, doi:10.3390/ijms26093963_

Round 1
Reviewer 1 Report
Comments and Suggestions for Authors
In this work by Tal Raviv and collaborators, the authors evaluate the mechanism by which resveratrol induces cancer cell death through the modulation of VDAC1 and its interaction with Hexokinase I. The results align with previous reports demonstrating the regulation of VDAC1 by phenolic compounds and are further strengthened by the extensive prior research of the authors on this topic. The study employs multiple experimental approaches, presenting coherent, consistent, and logically structured findings.
One of the most noteworthy findings, in my opinion, is the resveratrol-induced overexpression of VDAC1, given its implications for cancer cell biology. This raises several relevant questions worth addressing, such as whether other VDAC isoforms or Hexokinase variants are similarly affected by resveratrol. Additionally, what are the broader implications of VDAC1 modulation by resveratrol in more complex therapeutic contexts, including co-administration with other phenolic compounds targeting the same protein or in combination with VDAC-based peptides?
Line 26: "calcium" should be specified as calcium (Ca²⁺). Line 27: Please specify the Hexokinase isoform studied. Figure 1A: The cancer cell line used is not clearly indicated. Please specify this with greater clarity. For every figure, please state the statistical analysis used.
For calcium measurements, the authors should explicitly describe the gating strategies employed. The measured calcium levels were performed only in viable cells, correct? Since dying or dead cells often exhibit high autofluorescence, it is essential to clarify how this artifact was avoided to ensure that the observed fluorescence increase is attributed to calcium deregulation.
Figure 3D, E: The corresponding statistical evaluation is missing.
Figure 4: Although HKII has traditionally been considered more relevant in cancer tissues, the authors demonstrate that VDAC1 interactions are not limited to this isoform. However, the study focuses on HKI rather than HKII. Could the authors provide context for this choice? Is HKI more relevant than HKII in this specific setting, or was this decision based on other factors?
Figure 4: The results indicate a more diffuse distribution, but there is also a notable increase in HKI expression. Would the authors like to discuss the significance of this observation?
Author Response
As our response includes some figures, it is attached as PDF

Reviewer 2 Report
Comments and Suggestions for Authors
In the manuscript entitled "Resveratrol’s pro-apoptotic effects in cancer are mediated through interaction and modulation of the mitochondrial VDAC1," the authors investigated the pro-apoptotic activity of natural compound resveratrol in cancer cells. The authors claimed that they found VDAC1 to be a novel target for resveratrol. The authors reported that resveratrol (1) increased VDAC1 expression levels, (2) directly bound to VDAC1 and promoted VDAC1 oligomerization, (3) induced apoptotic cell death, (4) elevated intracellular calcium levels, (5) enhanced the production of ROS, (6) induced the detachment of HK-I (a key enzyme in mitochondrial metabolism) from VDAC1. They also found that VDAC1 siRNA knockdown reduced resveratrol-induced cell death, confirming the participation of VDAC1 in resveratrol-induced cell death.
This manuscript addressed the mechanism of pro-apoptotic effects of resveratrol, a commercially available dietary supplement. Although this activity is known (PMID: 36430164), the authors used experimental data to demonstrate the important role of VDAC1 expression and VDAC1 oligomerization in resveratrol-induced apoptosis. Their data also revealed the involvement of several mitochondrial factors in this apoptosis, including calcium levels, ROS, SIRT1, and HK-I. Overall, the research topic is important and interesting, and the manuscript has a good structure and flows logically. However, the authors should address the following concerns.
- (Line 2-4) I would suggest adding "oligomerization" to the title, as it is the main finding in this study.
- (Line 21-24) VDAC1 has been reported as a target of resveratrol (PMID: 38819635). It is inappropriate for the authors to describe this as their novel finding.
- (Line 598-607) In the methodology, the authors analyzed (1) apoptosis by Annexin V/PI staining and (2) cell death by PI staining. These two methods seem redundant, please explain why two similar methods are used.
- (Figure 1A) In the Annexin V/PI staining, resveratrol primarily increased dead cell populations in the PI+/AV- and PI+/AV+ quadrants. This pattern differs from classical apoptosis, which increases the PI-/AV+ (early apoptosis) and PI+/AV+ (late apoptosis) cell populations. Please explain the data interpretation.
- (Figure 1B) Please describe how to calculate the "apoptosis, %"
- (Figure 1B) The data showed that cancer cells SH-SY5Y are more sensitive to resveratrol-induced apoptosis than the non-cancerous HEK-293 cells. Are HEK-293 cells under resveratrol treatment different from SH-SY5Y cells in any of VDAC1 expression, VDAC oligomerization, calcium levels, ROS levels, or HK-I detachment?
- (Figure 1C) If the "cell death, %" was calculated by the PI+ cells in flow cytometry analysis, please explain why Annexin V was not used instead. Annexin V is a better marker for apoptosis as it detects early and late apoptosis, whereas PI can only detect late apoptosis.
- (Figure 1D) Please explain how the numbers of IC50 and maximal cell death were calculated.
- (Figure 2E) Marker mislabeling, two 25.
- (Figure 2E) Around 25-35 kDa, there are three bands; are they all VDAC1 monomers? Why does the 75 uM sample miss the lowest band?
Author Response
Reviewer 2
We thank this reviewer for the valuable feedback and comments, which we have addressed as presented below: our manuscript.
- Line 2-4) I would suggest adding "oligomerization" to the title, as it is the main finding in this study.
As suggested, Oligomerization has been added to the title.
2. (Line 21-24) VDAC1 has been reported as a target of resveratrol (PMID: 38819635). It is inappropriate for the authors to describe this as their novel finding.
The paper mentioned above, as noted in the Discussion (Lines 493-497), demonstrated that resveratrol's suppressed VDAC1 expression, reduced mtDNA release, inhibited mPTP opening, prevented mitochondrial dysfunction helped mitigate dopaminergic neuron degeneration and pathological progression in a Parkinson's disease model were obtained at low concentrations of resveratrol 10 and 40 µM. However, in our study we used high concentrations of resveratrol (100-100 uM) resulted in VDAC1 overexpression, oligomerization, and apoptosis. Moreover, we demonstrated the direct interaction of resveratrol with purified VDAC1. These suggest that our findings are novel findings.
The observed differences are due to the varying concentrations used—10 and 40 µM in the previous study compared to over 100 µM in our study. This is consistent with the dual effects of resveratrol: at low concentrations, it exhibits antioxidant, anti-inflammatory, neuroprotective, and antiviral properties, as well as improvements in cardio-metabolic health and anti-aging benefits, while at high concentrations, it induces cell death and inhibits tumor growth.
To emphasize these differences and suggest that VDAC1 plays a role in the concentration-dependent dual effects of resveratrol—pro-survival at low concentrations and pro-cell death at high concentrations—we add the following:
A recent study (90) demonstrated that VDAC1 is overexpressed in PD-like pathology, such as in A53T transgenic mice or cells exposed to MPP+ (1-Methyl-4-phenylpyridine), both of which induce PD-like conditions. At both conditions, resveratrol, at low concentrations (10-40 µM), decreased the overexpression of VDAC1, reduced mtDNA release (which is mediated by VDAC1 oligomers (87), inhibited mPTP opening, and prevented mitochondrial dysfunction. These effects helped mitigate dopaminergic neuron degeneration and pathological progression in a Parkinson's disease model (90).
These findings support the involvement of VDAC1 in the protective effects of resveratrol at low concentrations, including its antioxidant, anti-inflammatory, neuroprotective properties, as well as improvements in cardio-metabolic health and anti-aging effects[14-19] and at high concentrations, promote cell death and acts as an anti-cancer agent [9-12].
3. (Line 598-607) In the methodology, the authors analyzed (1) apoptosis by Annexin V/PI staining and (2) cell death by PI staining. These two methods seem redundant, please explain why two similar methods are used.
Annexin staining is used to detect apoptosis (early and late) and PI stains late stage of apoptosis and/or necrosis.
4. (Figure 1A) In the Annexin V/PI staining, resveratrol primarily increased dead cell populations in the PI+/AV- and PI+/AV+ quadrants. This pattern differs from classical apoptosis, which increases the PI-/AV+ (early apoptosis) and PI+/AV+ (late apoptosis) cell populations. Please explain the data interpretation.
The ratio between the four cell populations—live, Annexin V, Annexin V+ PI, and PI—depends on both the concentration of resveratrol and the incubation time. At the concentrations used in this study, we predominantly observed a late apoptotic state (Fig. 1).
5. (Figure 1B) Please describe how to calculate the "apoptosis, %"
As shown in Fig. 1A, we observed about 8% cell death in the control. However, in Fig. 1B, cell death in the control is presented as zero, suggesting that the PI-stained cells were not included in the apoptosis values presented in Fig. 1B.
6. (Figure 1B) The data showed that cancer cells SH-SY5Y are more sensitive to resveratrol-induced apoptosis than the non-cancerous HEK-293 cells. Are HEK-293 cells under resveratrol treatment different from SH-SY5Y cells in any of VDAC1 expression, VDAC oligomerization, calcium levels, ROS levels, or HK-I detachment?
HEK-293 cells are immortalized human embryonic kidney cells, which are considered non-cancerous, but have been reported to be tumorigenic, were presented as well Hela cells to indicated different sensitivity of various cell lines to resveratrol. We focused on the well-defined cancer cells to demonstrate our proposed mechanism of VDAC1-mediated resveratrol-induced cell death
7. (Figure 1C) If the "cell death, %" was calculated by the PI+ cells in flow cytometry analysis, please explain why Annexin V was not used instead. Annexin V is a better marker for apoptosis as it detects early and late apoptosis, whereas PI can only detect late apoptosis.
We agree with the reviewer that Annexin V + PI staining is a better method for assessing apoptosis. However, as shown in Fig. 1A, the Annexin V-stained cells represent only a small fraction of the total cell death. Therefore, we believe that the results obtained using Annexin V + PI staining or PI staining alone would be very similar.
8. (Figure 1D) Please explain how the numbers of IC50 and maximal cell death were calculated.
Both values were derived from Figures 1B and 1C. The IC50 represents the concentration that resulted in 50% of the maximal cell death, while maximal cell death refers to the highest percentage of cell death induced by the highest concentration used.
9. (Figure 2E) Marker mislabeling, two 25.
Thank you, we changed the lower one to 15 kDa
10. (Figure 2E) Around 25-35 kDa, there are three bands; are they all VDAC1 monomers? Why does the 75 uM sample miss the lowest band?
The band below the major VDAC1 band is found in all samples regardless resveratrol treatment, it represents VDAC3 as the anti-VDAC1 antibodies used (Cat# ab15895) recognize both VDAC1 and VDAC3. The second minor band, as indicated (legend to Fig. 2 and lines 169-171) represents VDAC1 intramolecular crosslinked (Refs. 37,43)
